# Chimeric Antigen Receptor T-Cell Therapy in Acute Myeloid Leukemia: State of the Art and Recent Advances

**DOI:** 10.3390/cancers16010042

**Published:** 2023-12-20

**Authors:** Martina Canichella, Matteo Molica, Carla Mazzone, Paolo de Fabritiis

**Affiliations:** 1Hematology, St. Eugenio Hospital, ASL Roma2, 00144 Rome, Italy; cmazzone1@virgilio.it (C.M.); paolo.defabritiis@aslroma2.it (P.d.F.); 2Department of Hematology-Oncology, Azienda Ospedaliera Pugliese-Ciaccio, 88100 Catanzaro, Italy; molica@bce.uniroma1.it; 3Department of Biomedicina e Prevenzione, Tor Vergata University, 00133 Rome, Italy

**Keywords:** acute myeloid leukemia (AML), chimeric antigen receptor (CAR)-T-cells, cellular therapy, antigens, preclinical models, clinical trials

## Abstract

**Simple Summary:**

Compared to the resounding success demonstrated in the field of B-cell leukemia, lymphoma, and multiple myeloma, in the field of acute myeloid leukemia, CAR-T-cell-therapy slows down its application in clinical practice. Yet, immunotherapy and/or cell therapy could be curative in certain high-risk AML subtypes refractory to classical chemotherapy approaches. Several CAR-T constructs targeting different antigens have been tested and have shown promising results. This review illustrates the main results obtained with the use of CAR-T in AML.

**Abstract:**

Chimeric antigen receptors (CAR)-T-cell therapy represents the most important innovation in onco-hematology in recent years. The progress achieved in the management of complications and the latest generations of CAR-T-cells have made it possible to anticipate in second-line the indication of this type of treatment in large B-cell lymphoma. While some types of B-cell lymphomas and B-cell acute lymphoid leukemia have shown extremely promising results, the same cannot be said for myeloid leukemias—in particular, acute myeloid leukemia (AML), which would require innovative therapies more than any other blood disease. The heterogeneities of AML cells and the immunological complexity of the interactions between the bone marrow microenvironment and leukemia cells have been found to be major obstacles to the clinical development of CAR-T in AML. In this review, we report on the main results obtained in AML clinical trials, the preclinical studies testing potential CAR-T constructs, and future perspectives.

## 1. Introduction

Acute myeloid leukemia (AML) is a hematological malignancy characterized by the clonal proliferation of immature and non-functional myeloid cells in bone marrow and blood. It represents the second most common form of leukemia in adults [1]. AML can be considered a heterogeneous disease with different morphological, genetic, cytogenetic, and clinical characteristics. It can onset de novo after the evolution of a previous myeloid disease or after previous chemo or radiotherapy (so-called therapy-related AML).

Despite the high rate of complete remission (CR) achieved after first line therapy (about 60–70% of patients), the overall survival rate of refractory/relapsed (R/R) AML patients remains too low (25% at 5 years) [2]. For high-risk patients, the only therapeutic option is allogeneic hematopoietic stem cell transplantation (allo-HSCT), although it is not accessible for all patients, especially the elderly. Immunotherapy-based approaches promise higher survival rates due to greater positive responses, lower toxicity (which allows more patients to be fitter and transplant-eligible), and a lower rate of minimal residual disease (MRD). However, there are difficulties in obtaining these desired results, which can be traced back to three main causes: the lack of a specific AML surface antigen, tumor heterogeneity, and the complexity of the bone marrow microenvironment. In fact, it is well known that the stromal cells network may preserve AML cells from the damage of chemo or cellular therapy. In recent years, a novel form of cellular immunotherapy has been explored based on the engineering of autologous T-cells expressing a chimeric antigen receptor (CAR) against tumor antigens. In the hematological field, CAR-T-cell therapy had produced significant results in the treatment of R/R diffuse large B-cell lymphoma (DLBCL), mantle-cells lymphoma (MCL), follicular lymphoma, B-cell acute lymphoblastic leukemia (B-ALL), and R/R multiple myeloma (MM) [3,4,5,6,7,8,9,10,11,12,13]. Unfortunately, the immunological complexity of AML makes it difficult to apply CAR-T in AML patients, both in preclinical model trials and in clinical practice. For this review, we retrospectively searched PubMed databases and meeting abstracts for clinical and preclinical applications of CAR-T in AML.

## 2. The Structure of CARs

From the first CAR designed by Zelig Sahar et al. in 1989, the basic structure remained the same. It consists of four domains (Figure 1) [14]:Antigen recognition domainHinge domainTransmembrane domainIntracellular signal domain

**Figure 1 cancers-16-00042-f001:**
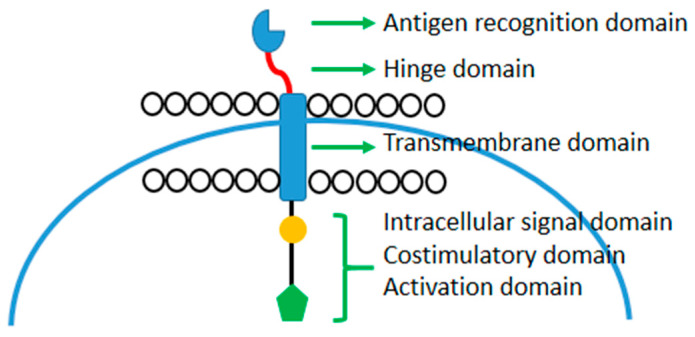
Basic structure of CAR-T.

### 2.1. Antigen Recognition Domain

The extracellular antigen recognition domain is fundamental for CAR-T-cell activation. The most common structure is similar to the single-chain variable fragment (skiff) monoclonal antibodies. It is composed of two chains the heavy (VH) and light (VL). Most innovative CAR constructs used antibody-derived binding components such as nanobodies. This part of the CAR receptor allows for antigen binding, antigen recognition, and cytotoxicity [15,16,17].

### 2.2. Hinge Domain

The hinge domain can be considered as a bridge between the extracellular (which connect the antigen recognition domain) and the transmembrane domains. Its main property is flexibility, which is responsible for CAR and expression, signal transduction, and epitope recognition. Currently, the most common hinge domains used for the CAR construct are represented by amino acid sequences derived from CD8, CD28, IgG1, or IgG4.

### 2.3. Transmembrane Domain

The transmembrane is probably the region less focused on, but this domain is also important for the correct functioning of CAR-T. The most common structures derived from natural proteins include CD3ζ, CD4, CD8α, or CD28. There is evidence about different characteristics of each of them. For example, CD3ζ facilitates CAR dimerization and activation through the incorporation of the endogenous TCRs [17,18]. Indeed, it seems that the type of transmembrane domains influence CAR-T-cell cytokine production. CD8a induces less IFNγ and TNFα release than CD28 and the activation-induced cell death (AICD) [19].

### 2.4. Intracellular Signal Domain

The function of the intracellular signal domain is the activation of T-cell engineer. This is possible with the combination of the activation domain and one or more co-stimulatory factors. The CD3z is the template of the activation part, but it is not sufficient to promote the persistence and activity of CAR-T [20].

The addition of co-stimulatory domains resulted in the better performance of CAR-T [21]. CD28 and 4-1BB are the two most utilized costimulatory domains used in the development of CAR-T approved by the FDA and EMA [3,4,5,6,7,8]. Currently, other costimulatory molecules are being tested in preclinical model KIR2DS2 (killer cell immunoglobulin-like receptor 2DS2), ICOS (inducible T-cell co-stimulator), CD27, MYD88-CD40, and OX40 (CD134) [22,23,24,25,26,27].

## 3. CAR-T Generations

To date, research has led to the development of five generations of CAR-T-cells based on variations in co-stimulatory molecules (Figure 2).

The first generation included CAR with CD3ζ without co-stimulatory domains. This construct failed in the persistence and activity of CAR-T-cells in vivo [24]. The second generation, as cited above, presented the additional co-stimulatory domains CD28 and 4-1BB, and they represented the CAR-T currently approved for clinical practice [25]. Then, to improve safety, a third generation was developed with the addition of multiple co-stimulatory domains. However, this does not translate to a significant success compared with the second generation. [26]. The fourth generation CAR is represented by T-cells redirected for universal cytokine-mediated killing (TRUCKs) with an added IL-12. IL-12 is expressed either constitutively or inducible after CAR activation, and support the elimination of malignant cells through mechanisms [27,28]. Finally, the fifth generation CAR was designed to involve the IL-2 receptor in the structure. This construct can activate three pathways (TCR, CD28 and JAK-STAT), which enhance the expansion of CAR-T [29,30,31].

## 4. CAR-T in Clinical Phase in AML

In theory, the ideal target antigen for CAR-T-cell therapy should have four principal characteristics:Expression in the majority of patientsExpression on blasts and on immature leukemia initiating clonesStable expression without downregulationAbsent on normal hematopoietic stem cells (HSC) and other normal tissue to avoid off-target effects

So far, no antigen on AML cells fulfills all criteria. Researchers have focused on several tumor antigens, but only a minority of them have resulted in effective targets for CAR. In Table 1, we report the clinical trial currently open to recruitment, while in Table 2, we report the results of the mail trials that have been published [32]. Finally, Table 3 summarizes the main AML target antigens used in clinical trials and in preclinical models.

### 4.1. CD33-CAR T-Cell

CD33 is a transmembrane receptor belonging to the sialic-acid-binding immunoglobulin-like lectin (Siglec) family. It is expressed both on myeloid progenitors and on differentiated cells [33]. About 90% of AML leukemic cells (both stem cells and blasts) express CD33, making it a suitable target [33]. Regarding the anti-CD33 CAR-T experience, after the first publication by Wang et al., different researchers have focused on the in vitro and in vivo efficacy of this construct [33]. Kenderian et al. developed a CD33 CAR (CART33) using the scFv of gentuzumab-ozogamicin. CART33 exhibited significant cytotoxicity in vitro and in xenograft model. However, the most common adverse effect was the irreversible aplasia [56]. To overcome the myelosuppression after CAR-T therapy, Minagawa et al. designed selectively inducible Caspase-9-CAR-T-cells which could specifically lyse CD33+ AML cells in vitro, largely eliminating CAR-T-cells by suicide gene activation [40]. Then, Li et al. demonstrated that 4-1BB as a co-stimulator domain increased central memory and prolonged survival compared with CD28 [57]. Subsequently, Zheng et al. used PI3K inhibitors to modulate the differentiation of CD33 CAR-T [58]. The history of CD33 CAR-T continues with Tambaro et al., which in 2011 conducted a phase I clinical trial to evaluate the efficacy and safety of CD33 CAR-T in R/R AML [59]. Then, in 2022, Liu et al. demonstrated that the third generation CD33 CAR-T exhibited stronger vitality, proliferation, and cytotoxicity than the second-generation [60]. However, the efficacy of CD33 CAR-T in vitro did not translate to promising results in clinical trials. More efforts are needed to improve the anti-leukemia response of CD33 CAR T in AML patients, and hopefully, we may see positive results in the ongoing clinical trials.

### 4.2. CD123-CAR T-Cell

CD123 is an IL-3 receptor alpha chain that acts as is normally expressed on HSC [44,45,61,62]. Indeed, this antigen is expressed in about 97% of AML patients and in 45% of these patients, it is overexpressed. Low levels of CD123 were observed on monocytes, dendritic cells, and endothelial cells [63]. In vitro and preclinical evidence demonstrated that CD123 CAR-T was effective against AML blasts without eliminating granulocytes/macrophages and erythroid colony formation [64]. However, Gill et al. demonstrated that the efficacy of CD123 CAR-T is associated with myeloablation in a xenograft mouse model, suggesting that CD123 CAR-based treatment may be used as a bridge for HSCT [65]. The first clinical application of CAR123 confirmed its anti-leukemic activity; however, two major issues emerged [66]. The first was the need to allograft post CAR123 treatment due to myeloablation. The second was the scarce persistence of CAR products, which may lead to AML progression. To address these challenges, different strategies were designed, including (1) transient activation of anti-CD123 with a messenger RNA electroporated platform; (2) T-cell depletion with alemtuzumab (monoclonal antibody against CD52) after CAR-T-cell therapy; and (3) T-cell ablation with rituximab to CD20-coexpressing CART123 (CART123-CD20) [67,68]. Indeed, Arcangeli et al. introduced a mutation in the anti-CD123 CAR antigen binding gene to decrease the affinity of CAR and minimize its toxicity and side effects [35,69]. These approaches together seem to be promising, and several clinical trials are currently exploring the feasibility of CAR123. Meanwhile, in recent years, CD123 seems to be a more effective target when used in an allogeneic CAR-T approach (see below).

### 4.3. CLL1-CAR T-Cell

C-type lectin-like molecule-1 (CLL-1) is a type II transmembrane glycoprotein expressed in about 92% of AML blasts and leukemia stem cells (LSCs), and absent in normal CD34+CD38-HSCs [70,71]. This expression pattern makes it a potential target for therapeutic intervention. The first in vitro experiments of CLL1-CAR by Tashiro et al. demonstrated a high efficacy in killing AML blasts without damaging healthy HSCs, allowing immune recovery after therapy [72]. The first report on the use of CLL1-CAR in an AML pediatric patient was by Liu et al. Compound CAR-T targeting CLL1 and CD33 was constructed, and alemtuzumab was administered with the goal of clearing CAR-T-cells after tumor eradiation. The patient received two split doses of CAR-T and achieved CR on day 19 [46]. Subsequent research focused on designing a different CLL1 CAR construct with better proliferation, functional persistence, and antitumor activity. Atilla and colleagues found that CD28ζ CAR with a short hinge region or with a CD8 intracellular domain produced better results than other CLL1 CARs [73]. They also demonstrated that CLL1 CAR with additional transgenic IL15 supplementation and a Caspase-9 control switch showed expansion, persistence, and anti-leukemia activity without excessive cytokine production. Further observations showed that PD-1 expression increased after the activation of CAR-T-cells and caused T-cell exhaustion. To overcome this issue, Lin et al. designed a PD1-silenced CLL-1 CAR and observed the same cytotoxicity with longer persistence [74]. Overall, CLL1-CAR shows promising anti-tumor efficiency in pre-clinical experiments and anti-AML cytotoxicity in spared normal HSCs [75].

### 4.4. CD70-CAR T-Cell

CD70, a member of the TNF-alpha family, represents an optimal target against AML because it is absent on HSCs, but highly expressed on myeloid blasts and on LSCs [76,77]. The history of CD70 CAR began with the observation by Reither et al. that the CD70/CD27 axis promotes the stemness of AML blasts, and at the same time, they demonstrated that a hypomethylating agent (HMA) upregulated CD70 expression in LSCs [41]. This evidence led to a phase I trial combining cusatuzumab (anti-CD70 monoclonal antibody) with azacytidine. Although this study reported a high CR rate, it was not confirmed in the subsequent phase II trial. However, these experiments opened the way to the development of CD70-specific CAR-T [78]. Sauer et al. designed the first generation of CD70-CAR T (full length-CD27), which proved effective against AML blasts [79]. Wang et al. subsequently developed a second-generation CAR that included a 4-1BB costimulatory domain and CD3ζ, which together conferred the highest IFNγ production [51]. More recently, Leick et al. focused on the hinge region [52]. Based on this evidence from the combination of AZA with cusatuzumab, AZA may also be used as a bridge in the treatment of patients with R/R AML to CD70-CAR T-cells. However, the safety and efficacy of CD70 CAR needs more evidence in clinical trials.

### 4.5. NKG2D Ligands-CAR T-cell

NKG2D is an activator receptor with a homodimer and hexamer form expressed on CD8+ T-cells, a small amount of CD4+ cells, γδ T-cells, NK cells, and NKT-cells. NKG2D has a wide spectrum of ligands that are overexpressed encountering DNA damage, infections, and malignant transformations, namely MICA, MICB, and UL16-binding proteins (ULBP) [80,81]. Therefore, NKG2D ligands in CAR-T-cells have the potential for applications. The first evidence of NKG2D ligands CAR efficacy was derived from murine studies by Sentman and colleagues, which demonstrated the eradication of MM, lymphoma, and ovarian cancers [82]. Baumeiste et al. conducted a phase I clinical trial with first-generation NKG2D-CD3ζ-CAR-T in AML and MDS patients [37]. One AML patient exhibited a high level of IFN-γ without any high-grade toxicities. There are no on-going clinical trials with NKGD2-CAR T [83]; however, further studies will be required to prove the efficacy and safety of NKGD2 CAR-T.

### 4.6. FLT3-CAR T-Cell

FMS-like tyrosine kinase 3 (FLT3) is a member of the class III receptor tyrosine kinases, mainly expressed on the cell surface of hematopoietic progenitor cells, and plays an important role in normal hematopoiesis such as proliferation, differentiation, and survival [84,85]. However, about 30% of AML have FLT3 gene mutations (internal tandem duplications (ITD) and alterations in the tyrosine kinase domain (TKD)) [86]. However, the ectodomain of the FLT3 molecule is usually spared. Indeed, hematopoietic cells expressed FLT3 at low levels, making it a suitable marker for immunotherapy. In a preclinical study, a second-generation FLT3 4-1BB-CD3-CAR T-cell demonstrated the ability to recognize and eliminate FLT3+ tumor cells. At the same time, no toxicity assessment in a xenograft model was observed [87]. In another preclinical study, Wang et al. demonstrated the ability of FLT3L-4-1BB-CD3ζ-CAR T-cells to target FLT3L+ leukemia cells [88]. This experiment confirmed the previous results with less off-tumor toxicity on healthy progenitors and HSCs.

### 4.7. CD7-CAR T-Cell

CD7 is a transmembrane glycoprotein expressed in T-cells, NK cells with a co-stimulatory role in B- and T-cell lymphoid interactions. CD7 is also expressed in around 30% of AML patients, but not in healthy myeloid cells. For engineered CAR T-cells, CD7 removal is mandatory to limit T-cell fratricide due to CD7 expression by T-cells [89]. Gomes-Silva et al., developed a CD7-CAR T-cell, which was engineered against CD7+ tumor cells using the CRISPR/Cas9 technique to remove the CD7 gene [90]. Subsequently, a second generation of CD7-knockout (CD7KO) CD28- CD3ζ- CD7-CAR T-cell was designed. The results showed high cytolytic effects against AML, and surprising elimination of primary AML blasts and leukemia colony-forming cells, sparing healthy and erythroid cells. These results suggest that CD7-CAR T-cells are a potent weapon for refractory or relapsed AML.

## 5. Potential CAR-T in Preclinical Phase in AML

In the field of AML research, there is a constant effort to identify novel target antigens and corresponding CAR-binding domains with potential therapeutic utility. The major challenge is to identify an antigen with no or low expression in HSCs to avoid CAR T-related myelosuppression. Because there is often an overlap of antigen expression between HSCTs and blasts, different mechanisms have been adopted to overcome this limitation. In recent years, different CAR-T-cells against promising antigens have been designed, and these have demonstrated efficacy in in vitro and in vivo models; however, the majority have not yet been tested in clinical trials.

### 5.1. FRβ-CAR T-Cell

The folate receptor (FR) family is a group of folate-binding protein receptors including four members (α, β, γ and δ) with different tissue distributions. In particular, FRα and FRβ are commonly upregulated in ovarian cancer and AML, respectively. FRα-specific CAR-T-cells were developed more than 20 years ago, and currently, 4-1BB co-stimulated CAR-T are being evaluated in clinical trials in patients with ovarian cancer [91]. FRβ is an attractive target because it is expressed on about 70% of primary AML blasts and is limited in normal tissue. Preclinical models demonstrate that FRβ can be upregulated by all-trans retinoic acid (ATRA), a drug approved for the treatment of acute promyelocytic leukemia. Given these unique characteristics, Lynn and colleagues developed and characterized FRβ-specific CAR constructs containing the m909 scFv which had been previously validated for recognition of human FRβ [47]. They found that m909-CAR T activation directly correlated with the expression of FRβ; in the presence of low FRβ levels, CAR activity was decreased as demonstrated by the reduction of IFNγ and cytolysis. In contrast, exposure of FRβ+ AML to ATRA upregulates the FRβ expression, resulting in higher IFNγ production and increased cytolytic activity. Interestingly, the ATRA-mediated induction of FRβ did not impact FRβ expression in healthy HSCT and m909 CAR T-cells, limiting the off-target effect. FRβ is a promising target for CAR-T-cell therapy, which may be augmented in combination with ATRA.

### 5.2. h8F4-CAR T-Cell

It is well known that in AML blasts, the PR1 peptide derived from the leukemia-associated antigen proteinase 3 is overexpressed in HLA-A2 cells. Molldream et al. observed that in vitro PR1-induced cytotoxic T lymphocytes (CTLs) recognized myeloid blasts [92]. Subsequently, Ma et al. developed an anti-PR1/HLA-A2-second generation CAR-T-cell using ScFv derived from an anti-PR1/HLA-A2 T-cell receptor (TCR)-like antibody, called h8F4 [93]. This PR1-CAR-T exhibited high cytotoxicity against AML blasts in vitro while sparing normal hematopoietic progenitors. These results suggest the potential of endogenous self-antigens for targeting by CAR-T-cells in AML patients; however, clinical tests are needed to confirm their effectiveness and safety profile.

### 5.3. WT1-CAR T-Cell

Wilms Tumor 1 (WT1) is an oncogenic, zinc-finger transcription factor. WT1 is important for different cellular processes such as organ development, differentiation, and apoptosis. Its expression is low in bone marrow, the kidney, gonads, and spleen, and it is overexpressed in various hematological malignancies (AML and CLL) and in several solid tumors (glioblastoma, mesothelioma and ovarian cancer) [39,94,95]. Rafiq et al. developed and tested WT1-CAR-T-cells as another TCR mimic CAR using HLA-A*02:01, a peptide that arose from the WT1 antigen [48]. The efficacy of this construct has been demonstrated in the lysis activity and secretion of IL-2 and IFN-γ against cell lines in an in vitro model. Further experiments and in vivo studies are necessary to confirm the robustness of this target.

### 5.4. LILRB4-CAR T-Cell

LILRB4, a leukocytic, immunoglobulin-like receptor-B family, is a potential target antigen in AML. In a preclinical experiment, 41BB-CD3ζ-anti-LILRB4-CAR-T was engineered using the humanized ScFv to specifically target LILRB4 AML cells. The results were interesting due to the potent cytotoxic activity against AML blasts, together with a reduction in off-tumor toxicity by sparing normal progenitors and HSCs [96].

### 5.5. CD84-CAR T-Cell

A potential new target in the development of CAR T-cells in R/R AML is CD84, an immunoreceptor member of SLAM family (SLAMF5). Similar to CD123, CD84 is overexpressed on the surface of AML cells, but is low in CD34+ HSC and absent in other tissue. These features make CD84 a suitable target to design CAR-T. Pèrez-Amill et al. developed a CD84 CAR T using both human and murine CD84-binding sequences, resulting in six different constructs [97]. In vitro experiments demonstrated cytotoxicity against an AML cell line while sparing healthy HSCs. These results were confirmed using two of the six CD84-CAR-T in a NSG xenograft mouse model. CD84 CAR-T has the potential to become an efficient and safe treatment for AML.

### 5.6. Siglec-6-CAR T-Cell

Siglec-6 is expressed in myeloid leukemic blasts, but is absent on normal HSC cells. Siglec-6 consists of three extracellular immunoglobulin (Ig) domains and two intracellular immunoreceptor tyrosine-based inhibition motifs (ITIMs). ITIM motifs are fundamental for regulating and activating functions in immune cells [49,55]. Siglec-6 is also expressed in B-cells, mast cells, and placenta cells, but are absent on T-cells, natural killer (NK) cells, neutrophils, macrophages, and monocytes. In AML, it is expressed in primary AML blasts and absent in normal HSCs. Jetani et al. developed a Singlec-6 CAR T with a targeting domain derived from the mAb JML-1 [98]. In vitro and in vivo experiments revealed a high effectiveness against AML blasts, without impairing normal HSC activity. These preliminary results suggest that Siglec-6 CAR-T may be an important novel potent target in AML treatment; however, it requires much more investigation.

### 5.7. TIM-3 CAR T

T-cell immunoglobulin mucin-3 (TIM-3) is an immune checkpoint protein expressed on the membrane of leukemia stem cells, which can inhibit cancer immunity [99]. TIM-3 as a surface molecule is expressed on LSCs in almost all types of AML, but not on HSCs. The injection of TIM-3(+) AML cells into immunodeficient mice led to AML reconstitution, and these data—plus the observation that TIM-3 is not expressed in normal tissues and HSCs—make it an attractive target. Utilizing scFv from a TIM antibody, a second generation TIM3 CAR-T was developed. In xenograft models, TIM3 CAR-T showed high antileukemia activity with the production of IFN-γ, granzyme B, and perforin. Xin He et al. demonstrated in a mouse model the efficacy and low toxicity of CAR-T-cells, specifically targeting both CD13 and TIM3 that are upregulated in LSCs [100]. Further experiments on TIM-3 could lead to the development of an effective AML CAR treatment.

### 5.8. CD93 CAR-T-Cell

CD93 is a C-TYPE lectin transmembrane receptor, and it can be considered an adhesion molecule. Among AML antigen markers used to monitor minimal residual disease, CD93 showed an interesting expression profile due to its presence in AML blasts and its absence in other immune cells [42,50]. Richards et al. designed a CD93 CAR-T using a humanized CD93-specific binder [101]. Their results showed strong anti-leukemic activity while sparing healthy HSCs. However, the presence of CD93 on the surface of endothelial cells led to an important “off-target” toxicity. To address the challenge of endothelial-specific cross-reactivity, a model based on NOT-gated CD93-CAR T-cells was developed that circumvented endothelial cells. In this model, endothelial cells were protected and not damaged from CD93 CAR T. CD93 represents a potentially effective target for CAR-T in AML treatment; however, the cross-reactivity against endothelial cells is a challenging problem to overcome [101].

## 6. CAR-T-Cells Side-Effects

The success of CAR-T-cell therapy also depends on the management of the specific toxicities which are illustrated below, and on the strategy to overcome CAR-T limitations (Section 7). In these two sections, we reported the consolidated date regarding the toxicity derived from the pivotal CAR-T trials in which NHL, LAL-B and MM were treated in patients. However, as shown in Table 3, the most common side effects in the treatment of AML patients with CAR-T are represented by cytokine release syndrome (CRS) and immune-effector-cell-associated neurotoxicity syndrome (ICANS). Indeed, in the case of CAR-T constructs that are expressed both on blast cells and on HSCT, myelosuppression was observed.

### 6.1. Cytokine Release Syndrome (CRS)

Considering all grades, around 77–93% of leukemia patients and 37–93% of lymphoma patients receiving CAR-T-cell therapy manifested CRS [3,4,5,6,7,8,9,10,11,12]. CRS is triggered by inflammatory cytokines and chemokines released by CAR-T-cells, such as interferon (IFN)γ, tumor necrosis factor (TNF)α, granulocyte macrophage colony stimulating factor (GM-CSF), and interleukins (IL)-2, IL-8, and IL-10 [98,99]. Xenogeneic models demonstrated that CRS is a supramaximal response of host immune cells rather than CAR T-cells, and demonstrated that the main actor of CRS is IL-6. The main symptoms of CRS are constitutional, such as fever associated with fatigue, myalgia, arthralgia, rigor, and/or anorexia. In some cases, these symptoms could rapidly deteriorate into hypotension, tachycardia, tachypnea, hypoxia, arrhythmia, capillary leak, coagulopathy, respiratory failure, shock, and/or organ dysfunction. The median time to CRS onset varies and depends on the specific CAR T-cell product. For the 4-1BB (CD137) costimulatory domain, the CRS risk peaks at 3 and 7 days after CAR T-cell administration for ALL and DLBCL patients, respectively. In contrast, the onset of CRS symptoms in patients treated with axicabtagene ciloleucel or brexucabtagene autoleucel containing CD28 as a costimulatory domain is earlier, usually 2 days after CAR T-cell administration. However, CRS can occur up to 3 weeks after CAR T-cell administration [102,103,104].

To standardize CRS grading and toxicity management, different multi-institutional scoring systems have been developed. Table 4 reported ASTCT grading [105]. According to these guidelines, the appearance of a fever—defined as a temperature of 38 °C or greater—within 24 h to 3 weeks after CAR T-cell administration is the prerequisite for CRS diagnosis. The grade and severity of CRS is based on hypotension and hypoxia. The administration of tocilizumab, a monoclonal antibody against IL-6, may be considered in patients with CRS grade 2. The early use of tocilizumab reduced the severity of CRS, sparing the function of CAR-T. The combination of corticosteroid may be considered if CRS presents within ICANS and/or the patient presents a deterioration after two doses of tocilizumab.

### 6.2. Immune-Effector-Cell-Associated Neurotoxicity Syndrome (ICANS)

Neurotoxicity following CAR-T-cell therapy is relatively common and can occur in up to 67% and 62% of leukemia and lymphoma patients, respectively [54]. High-grade ICANS is likely more frequent for CAR-constructs, with CD28 as a costimulatory domain, occurring in up to 45% of treated patients. In contrast, ICANS develops less frequently in patients treated with 4-1BB-containing CAR-T-cells, i.e., tisa-cell, with severe ICANS observed in up to 13% of patients. The pathophysiology of ICANS is less known, and several mechanisms have been identified: passive diffusion of cytokines, endothelial activation, and microglial activation with secretion of cytokines [106]. ICANS management is based on the administration of corticosteroids, but instead, the IL-6 inhibitors are often not effective for neurotoxicity associated with CAR-T-cell therapy [101]. Multiple groups have demonstrated the successful use of the IL-1 antagonist anakinra to ameliorate CRS in mouse models [107]. To date, there are no prophylaxis for the prevention of CAR-T neurotoxicity, so it is mandatory to optimize CAR engineering and develop strategies to decrease CAR-induced ICANS. Table 5 illustrates ASTCT ICANS grading [53,108].

### 6.3. Hematological Toxicity

In the pivotal studies ZUMA-1 and JULIET, around 31% and 17% of patients, respectively, presented grade ≥ 3 febrile neutropenia. Indeed, 30% of patients treated with axi-cel or tisacel beyond 30 days presented with severe cytopenia [43,109].

CD19-CAR-T presented a biphasic pattern of early and late cytopenia.

Early cytopenia has been attributed to the lymphodepleting chemotherapy, while the etiology of late cytopenia is less clear.

Late cytopenia presents a multifactorial etiology; it may be associated with prior chemotherapy and/or allo-HSCT or a high grade of CRS. The treatment of anemia and thrombocytopenia is based on erythrocytes and platelets transfusions. Granulocyte colony-stimulating factor (G-CSF) can be used to treat prolonged neutropenia. If the cytopenia is persistent, it can be resolved by autologous or allogeneic stem cell supports [110,111].

## 7. CAR-T-Cell Limitations

Despite its remarkable effectiveness and durable clinical responses, the previous clinical trials in B-cell malignancies demonstrated the limitations of CAR-T-cell therapy, which could be applied in the fields of AML discussed belove. Future studies with a larger number of patients and longer follow-up still must be addressed, which should include life-threatening CAR-T-cell associated toxicity as discussed above, antigen escape, limited persistence, poor trafficking, tumor infiltration, and the immunosuppressive microenvironment. These are discussed below.

### 7.1. Antigen Escape

One of the biggest challenges of CAR-T-cell therapy is tumor resistance to single-antigen-targeting CAR constructs known as antigen escape. Recent data demonstrated that in about 30–70% of patients, the CD19 antigen was downregulated or completely lost after relapse. Similarly, downregulation or loss of BCMA expression is observed in MM [54,105]. To reduce the relapse rate after CAR-T-cell treatment, several strategies are under investigation based on targeting multiple antigens; among them is the use of dual CAR constructs or tandem CARs. In the first evidence concerning in adult patients with ALL and DLBCL, CD19/CD22 CAR-T has demonstrated promising results, such as in MM regarding the BCMA/CD19-targeted CARs. These data demonstrate the importance of optimizing target antigen selection to improve antitumor response and to decrease antigen escape mechanisms causing relapse [112,113,114,115,116].

### 7.2. On-Target Off-Tumor Effects

Antigen selection is crucial in CAR design, not only to optimize therapeutic efficacy, but also to limit “on-target off-tumor” toxicity. In hematological malignancies, the use of CAR-T anti-CD19 often results in B-cell aplasia. Further development of innovative strategies to reduce antigen escape, and for selection of antigens with sufficient antitumor activity and minimal toxicity, will be necessary to expand the clinical use of CAR-T-cell therapies in hematological malignancies.

## 8. Next-Generation CAR-T

As discussed above, current CAR-T-cell therapies approved in USA and Europe are based on second generation constructs, which present two main limitations. The first is their short persistence, which may reduce or nullify treatment efficacy. Pivotal clinical trials using these constructs revealed a relapse rate around 66%. The second aspect is the appearance of life-threatening adverse effects. In recent years, researchers have developed various next-generation CAR-T constructs based on advances in immunology and molecular engineering. Currently, four distinctive approaches are being tested in clinics:Modulation of immune checkpoint pathways (so-called armored CAR-T-cell).Induction of cytokine secretion (defined T-cells redirected for universal cytokine-mediated killing-TRUCKs).Implementation of a safety-switch mechanism to control the treatment-related adverse events.Allogeneic CAR-T (universal CAR-T -UCAR-T-).

The field of CAR-T-cell treatments is very promising, including the use of bispecific second-generation CAR-T-cells that simultaneously target two surface antigens. These studies have had positive results in preclinical and clinical studies with the CAR based on NK cells [117,118].

### 8.1. Armored CAR-T-Cells

To overcome the inhibition of the tumor microenvironment, different methods have been applied to disrupt the PD1-PDL-1 axis. Liu et al. studied 17 B-cell non-Hodgkin lymphoma patients treated with a CAR-T construct in which the extracellular PD-1 was fused to the intracellular CD28 activating domain, thus transforming the binding of PDL-1 to PD-1 into an activating signal [119]. The results were encouraging, although inferior compared with the rate of complete response of the pivotal studies. Another study disrupted PD-1 signaling by programming CAR-T-cells to secrete PD-1 Fc fusion proteins, preventing its suppressive effects on T-cells. No clinical results have yet been published. The most successful approach seems to be with the use of anbalcabtagene autoleucel (Anbal-cel). This is a novel CAR-T-cell product in which the PD-1 and T-cell immunoreceptors were silenced with Ig and ITIM (immunoreceptor tyrosine-based inhibitory motif) domains previously demonstrated to exhibit significant success in treating B-cell lymphoma [120]. Other approaches used in B-cell non-Hodgkin’s lymphoma include the knockdown PD-1 gene and the incorporation of cytosolic-activated PD-1 by the use of clustered regularly interspaced short palindromic repeats (CRISPR) technology. Armored CAR-T-cells appear to be a valid therapeutic approach in the treatment of both CD19 positive malignancies and myeloid leukemia.

### 8.2. TRUCK CAR-T

TRUCKs are next-generation CAR-T-cells engineered to express certain cytokines with the aim of improving CAR-T anti-tumor efficacy and persistence, and modulating the tumor microenvironment [112]. An unusual CAR-T construct was developed, incorporating the expression of interleukin-7 (IL-7) and chemokine (C-C Motif) Ligand 19 (CCL19) to mimic the cytokine environment in lymphoid organs. IL-7 promotes the proliferation and survival of T-cells, whereas CCL19 is a chemoattractant for both T-cells and dendritic cells (DCs). IL-7/CCL19-expressing CAR-T-cells were assessed in R/R DLBCL and MM patients, and showed encouraging preliminary results. Interestingly, the TRUCKS approach was also analyzed in patients with Hodgkin’s Lymphoma and cutaneous T-cell lymphoma in which the microenvironment plays a major role. In these settings, anti-CD30 CAR-T incorporating C-C chemokine receptor type 4 (CCR4) showed promising results [121]. These data suggest that TRUCK CAR-T-cells could be another weapon against myeloid leukemia and related diseases, although their efficacy and safety must be confirmed with clinical trials.

### 8.3. Switchable CAR-T-Cells

Switchable CAR-T-cells have been proposed to control related CRS and neurotoxicity. The rationale behind this approach is to induce apoptosis and complement-dependent cytotoxicity (CDC) and/or ADCC through the administration of an exogenous agent. The mechanism is based on the introduction in the CAR-T of genes encoding surface proteins/antigens or intracellular effectors. When this gene is expressed, CAR-T become responsive to specific drugs and stop their activity. An example of this is the incorporation of truncated epidermal growth factor receptor (EGFRt), into CAR-T-cells. The administration of monoclonal antibody cetuximab may target EGFRt, leading to the elimination of CAR-T reactives via ADCC. The results are still unpublished.

### 8.4. UCAR-T in AML

Another attractive and innovative option is the use of donor-derived CAR-T-cells, called allogeneic CAR-T, universal CAR-T (UCART), or off-the-shelf CAR-T. The advantages of using a product derived from a healthy donor are multiple. First, UCART is an immediately available product, overcoming the delay of manufacturing autologous CAR-T-cells (three to four weeks), which may be fatal for a highly aggressive disease. Second, the generation of a high number of CAR-T-cells from a single donor decreases overall production costs compared to that of autologous CAR-T-cells. For example, manufacturing UCAR-T-cells produces multiple batches of cryopreserved T-cells, allowing for the possibility of administering several CAR-T doses. Lastly, autologous T-cells may not be effective in some patients due to previous treatments. Despite these positive aspects, UCAR-T-cells present some limitations due to HLA disparities, such as the possibility of inducing graft-versus-host disease (GvHD) and host–mediated rejection. To avoid GvHD, different strategies have been developed. The most explored is genome editing of the T-cell receptor alpha constant domain (TRAC) through transcription activator-like effector nucleases (TALEN) technology [122,123,124]. TALEN has been utilized to disrupt the CD52 gene to reduce rejection. This approach suppresses all CD52+ cells with alemtuzumab, but makes allogeneic CAR-T-cells resistant to lymphodepletion, promoting their persistence. In a phase 1 trial, Sallman and colleagues tested UCART123v1.2, an anti-CD123 allogeneic CAR-T-cell product, in adult patients with relapsed or refractory AML. The results of this study was promising and support the use of UCART123 with fludarabine, cyclophosphamide (FC), and alemtuzumab, a regimen that demonstrated a significantly higher UCART123 cell expansion compared with FC [125].

## 9. Conclusions

In the past few decades, immunotherapy has emerged as the most effective treatment against solid tumors and hematological malignancies. The success achieved in the treatment of B-cell malignancies using immunotherapeutic approaches is not applicable in AML owing to the absence of a selective antigen on blast cells and the overlap of antigen expression between blasts and normal hematopoietic cells. Studies exploring multiple therapeutic targets are already underway, with promising results in preclinical and clinical studies. In addition, encouraging results in clinical trials on the treatment of AML have been reported, using targets such as CD123, CD33, CLL1, and CD70. The collective efforts of researchers and clinicians for developing novel CARs and applying them in current clinical trials will fulfill the promise to find effective cellular treatments for AML patients. At the same time, research must also be directed towards the development of technologies to limit toxicities and develop more efficient CAR-T constructs.

## Figures and Tables

**Figure 2 cancers-16-00042-f002:**
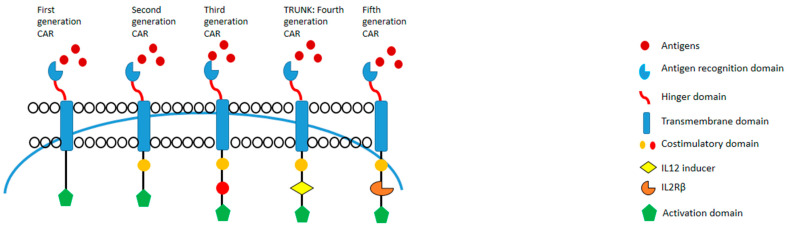
Five generations of CAR-T.

**Table 1 cancers-16-00042-t001:** Currently clinical trials open to recruitment.

Target Antigen	Clinical Trial ID	Phase	Disease Status	Intervetion	Institution
CD33	NCT05672147	Phase I	AML R/R	CD33 CAR-T	City of Hope Medical Cente
	NCT05248685	Phase I	AML R/R	CD33/CLL1 CAR-T	Beijing Boren Hospital
	NCT04835519	Phase I/II	AML R/R	CD33 CAR-T	Beijing Boren Hospital
	NCT04010877	Phase I/II	AML R/R	UCARTCD33 CLL-1, CD33, CD38 and CD123	Shenzhen Geno-Immune Medical Institute
	NCT03971799	Phase I/II	AML R/R	CD33 CAR-T	Center for International Blood and Marrow Transplant Research
	NCT05105152	Phase I	AML R/R	CD33 CAR-T	Seattle Children’s Hospital
CD123	NCT04265963	Phase I	AML R/R	CD123 CAR-T	Chongqing Precision Biotech Co., Ltd.
	NCT04272125	Phase I	AML R/R	CD123 CAR-T	Chongqing Precision Biotech Co., Ltd.
	NCT04010877	Phase I	AML R/R	CAR-T CLL-1, CD33 and/or CD123	Shenzhen Geno-Immune Medical Institute
	NCT05457010	Phase I	AML R/R	SPRX002 and ARC T	Arcellx, Inc.
	NCT04230265	Phase I	AML R/R and BPDCN	UniCAR02-T	AvenCell Europe GmbH
	NCT06125652	Phase I/II	AML R/R	anti Tim3/CD123	Xuzhou Medical University
CLL1	NCT04923919	Early Phase 1	AML R/R	CLL1 CAR-T	920th Hospital of Joint Logistics Support Force of People’s Liberation Army of China
	NCT05248685	Phase 1	AML R/R	Dual CD33/CLL1 CAR-T	Beijing Boren Hospital
	NCT04219163	Phase 1	AML R/R	CLL1 CAR-T	Baylor College of Medicine
	NCT04884984	Phase 1/2	AML R/R	CLL1 CAR-T	The First Affiliated Hospital of Soochow University
	NCT04010877	Phase 1/2	AML R/R	CLL1 CAR-T	Shenzhen Geno-Immune Medical Institute
	NCT06017258	Phase 1	AML R/R	CD371-YSNVZ-IL18 CAR-T	Memorial Sloan Kettering Cancer Center (Responsible Party)
	NCT04789408	Phase 1	AML R/R	CLL-1 CAR T	Kite, A Gilead Company
	NCT06110208	Early Phase 1	AML R/R	CLL1 and CD38 dual-target CAR-T injection	920th Hospital of Joint Logistics Support Force of People’s Liberation Army of China
CD70	NCT04662294	Early Phase 1	Non-Hodgkin’s Lymphoma Multiple Myeloma AML R/R	CD70 CAR-T	Zhejiang University
FLT3	NCT05023707	Phase 1/2	R/R AML	Anti-Flt3 CAR-T	The First Affiliated Hospital of Soochow University
	NCT05445011	Phase 1	R/R AML	Anti-Flt3 CAR-T	Wuhan Union Hospital, China
CD7	NCT04762485	Phase 1/2	R/R AML R/R T ALL	Anti-CD7 CAR-T	The First Affiliated Hospital of Soochow University
	NCT04033302	Phase 1/2	R/R AML R/R T ALL	Anti-CD7 CAR-T	Shenzhen Geno-Immune Medical Institute
	NCT05907603	Early Phase 1	R/R AML R/R T ALL	RD13-02 cell infusion	Kai Lin Xu, MD

**Table 2 cancers-16-00042-t002:** Results of main AML CAR-T clinical trials published.

Antigens	Pts (nr)	Lymphodepletation Regimen	Best Responce	Side Effects (Grade)			Ref
				CRS	ICANS	OTHER	
CD33	1	NA	1 PR	grade 2	-	Aplasia	[33]
	3	1 ARA-C 2 g/m^2^+ MTZ 8 g/m^2^/day 1 Flu30 mg/m^2^+ ARA-C 1 g/m^2^ 1 ARA-C 1.5 gr/m^2^ + MTZ 10 mg/m^2^	1 CR 2 PD	2 pts grade 2		3 pts Sepsis	[34]
CD123	7	Flu and Cy	1 CR i 2CR 1MLFS 1PD 2 SD	8 pts grade 1–3	4 pts grade 1–3	3 sepsis 9 aplasia	[35]
CLL-1	3	NA	3 CR 7 no response	2 pts grade 1–3			[36]
NKG2D	7	NA	7 no response	NA			[37]
	2	Flu 30 mg/m^2^ + Cy 300 mg/m^2^	1 SD 1 no response	2 pts grade 3			[38]

Abbreviations: PTs patients; nr numbers; CR complete remission; PR partial response; Ref references; CRS cytokine release syndrome; ICANS immune effector cell-associated neurotoxicity syndrome; NA not available; Flu fludarabine; Cy cyclophosphamide; MTZ mitoxantrone; MLFS morphologic leukemia-free state.

**Table 3 cancers-16-00042-t003:** AML target antigens.

Target Antigens	Type of Molecule	Role	On HSCs	On LSCs	On AML Blasts (% of AML Cases)	References
CD7	Ig superfamily/Glycoprotein	B and T-cell lymphoid development, transmembrane protein	No	Yes	Yes 30%	[39]
CD33	SIGLEC Family	Transmembrane receptor	Yes	Yes	Yes 90%	[40]
CD70	Member of the TNF-alpha family	Transmembrane protein	No	Yes	Yes ≥90%	[41]
CD84	Immunoreceptor member of SLAM family	Transmembrane receptor	No	Yes	Yes 99%	[42]
CD93	C-TYPE lectin transmembrane receptor	Cell-adhesion process	No	Yes	Yes	[43]
CD123	Type I cytokine receptor of IL-3	IL-3 receptor α subunit	Yes	Yes	Yes 97%	[44,45]
CLL1	Type II transmembrane glycoprotein	Transmembrane receptor	No	Yes	Yes 92%	[46]
FLT3	Type III cytokine receptor	Tyrosine kinase receptor	Yes	Yes	Yes 30%	[47]
FRβ	Folate-binding protein receptor	Folate delivery	No	Yes	Yes 70%	[48]
h8F4	Peptide derived from the leukemia-associated antigens proteinase 3	HLA-presented antigens	No	No	No ≥90%	[49]
LILRB4	Leukocyte Ig-like Receptor-B Family	Inhibitory receptor role in immune tolerance	No	Yes	Yes ≥90%	[50]
NKG2D	C-type lectin-like receptor Protein	Activator receptor	No	Yes	Yes NKG2D ligands 67–100%	[51,52]
Siglec-6	SIGLEC Family	Transmembrane receptor	No	No	Yes70%	[53]
TIM-3	T-cell immunoglobulin mucin-3	Immune checkpoint protein	No	Yes	Yes78%	[54]
WT1	Zinc-finger DNA binding Protein	Transcription factor	No	No	Yes 80–90%	[55]

Abbreviations: AML acute myeloid leukemia; HSCs hematopoietic stem cells; LSCs leukemic stem cells; SIGLEC sialic acid binding immunoglobulin-like lectin; TNF tumor necrosis factor; NKG2D natural killer group 2 D; FLT3 fms-like tyrosine kinase 3; FRβ folate receptor β; WT1 Wilms tumor 1.

**Table 4 cancers-16-00042-t004:** ASTCT CRS consensus grading.

CRS Parameter	Grade 1	Grade 2	Grade 3	Grade 4
Fever	Temperature ≥ 38 °C	Temperature ≥ 38 °C	Temperature ≥ 38 °C	Temperature ≥ 38 °C
Hypotension	None	Not requiring vasopressors	Requiring vasopressor with or without vasopressin	Requiring multiple vasopressors (excluding vasopressin)
			And/or	
Hypoxia	None	Requiring low-flow nasal cannula or blow-by	Requirinh high-flow nasal cannula, facemask, nonrebreather mask or Venturi mask	Require positive pressure (CPAP, BiPAP, intubation and mechanical ventilation)

**Table 5 cancers-16-00042-t005:** ASTCT ICANS consensus grading for adults.

Neurotoxicity Domain	Grade 1	Grade 2	Grade 3	Grade 4
ICE score *	7–9	3–6	0–2	0 (patient is unarousable and unable to perform ICE)
Depressed level of consciousness	Awakens spontaneously	Awakens to voice	Awakens only to tactile stimulus	Patient is unarousable or requires vigorous or repetitive tactile stimuli to arouse. Stupor or coma
Seizure	NA	NA	NA	Deep focal motor weakness such as hemiparesis or paraparesis
Motor findings	NA	NA	NA	Deep focal motor weakness such as hemiparesis or paraparesis
Elevated intracranial pressure/cerebral edema			Focal/local edema on neuroimaging	Diffuse cerebral edema on neuroimaging; decerebrate or decorticate posturing; or cranial nerve VI palsy; or papilledema; or Cushing’s triad

* Immune-effector-cell-associated encephalopathy (ICE) SCORE; Orientation: orientation to year, month, city, hospital: 4 points; Naming: ability to name three objects (e.g., point to clock, pen, button): 3 points; Following commands: ability to follow simple commands (e.g., “Show me two fingers” or “Close your eyes and stick out your tongue”): 1 point; Writing: ability to write a standard sentence (e.g., “Our national bird is the bald eagle”): 1 point; Attention: ability to count backwards from 100 by 10: 1 point; Scoring: 10, no impairment; 7–9, grade 1 ICANS; 3–6, grade 2 ICANS; 0–2, grade 3 ICANS; 0 due to patient unarousable and unable to perform ICE assessment, grade 4 ICANS.

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
