# Peer review of "Chimeric Antigen Receptor T-Cell Therapy in Acute Myeloid Leukemia: State of the Art and Recent Advances"

_cancers, 2023, doi:10.3390/cancers16010042_

Round 1

Reviewer 1 Report

Comments and Suggestions for Authors

The manuscript entitled “Chimeric antigen receptor T-cell therapy in acute myeloid leukemia: state of art and recent advances” provides a comprehensive review of the current research status and progress of CAR-T therapy in acute myeloid leukemia. However, several deficiencies should be improved in later revisions:

Major points:

1.     In the overview of research progress on NKG2D-CAR-T, the conceptual description of NKG2D is not sufficiently accurate. Actually, NKG2D, as the activating receptors on NK cells, is expressed in NK cells as well as many T cells, such as NKT cells, CD8+ T cells, and γδT cells. The ligands for NKG2D, also known as NKG2DLs, consist of MICA/B and ULBP1-6.

2.     The summary of recruiting AML clinical trials in Table 1 is not entirely accurate. For example, a clinical trial of NKG2D-base CAR against relapsed/refractory AML is recruiting (NCT 04167696).

Minor points:

1.     In Table 2, the references for the corresponding target antigens have not been fully cited. The detailed bibliographic information should be completed and filled in for this section.

2.     The sixth primary heading “CAR-T cells side-effects” is not bolded like the other primary headings and needs to be corrected for consistency

Comments on the Quality of English Language

Minor editing of English language required.

Author Response

1.In the overview of research progress on NKG2D-CAR-T, the conceptual description of NKG2D is not sufficiently accurate. Actually, NKG2D, as the activating receptors on NK cells, is expressed in NK cells as well as many T cells, such as NKT cells, CD8+ T cells, and γδT cells. The ligands for NKG2D, also known as NKG2DLs, consist of MICA/B and ULBP1-6.

R: We agree with the Reviewer. We have modified the description of NKGD2 to clarify the function of this receptor.

2. The summary of recruiting AML clinical trials in Table 1 is not entirely accurate. For example, a clinical trial of NKG2D-base CAR against relapsed/refractory AML is recruiting (NCT 04167696).

R: Table 1 refers to clinical trials still in active recruitment phase and the results of which are not yet available (we have clarified the rationale of the table in the text). Therefore, regarding NKGD2 ligands CAR-T there are one clinical trial not yet recruiting (NCT04658004 NKG2D CAR-T Cell Therapy for Patients with Relapsed and/or Refractory Acute Myeloid Leukemia) and one completed (NCT02203825). The study cited by Reviewer NCT04167696 does not concern NKGD2 ligands CAR-T (Study in Relapsed/Refractory Acute Myeloid Leukemia or Myelodysplastic Syndrome Patients to Determine the Recommended Dose of CYAD-02 (CYCLE-1).

3. In Table 2, the references for the corresponding target antigens have not been fully cited. The detailed bibliographic information should be completed and filled in for this section.

R: Thanks for the suggestion, we have made table 2 more complete.

4. The sixth primary heading “CAR-T cells side-effects” is not bolded like the other primary headings and needs to be corrected for consistency.

R: We corrected it.

Reviewer 2 Report

Comments and Suggestions for Authors

General remarks

1. No methodology: should be added

2. References

A lot of issues, incorrect references, absent references, incorrect references (eg name author misspelled), wrong way of referring. This should be thoroughly corrected. The number of mistakes was so high, that I had to stop keeping track of them.

3.  Incompleteness

The tabels listing clinical trials and target antigens are incomplete. Methodology used might clarify why.

4. Structure

Structure of part  4& 5 is not clear and should be adapted (see below in specific comments)

5. Spelling mistakes should be corrected, and article should be proofread for correct English

Specific remarks

1.     Title: state of the art

2.     Reference mistakes: eg reference 32->35 all wrong, do not match with what is referred to.

3.     P2, r3: spelling error: in fact (and not infact)

4.     P3, r3: spelling error: “There is evidence” not “There are evidence”

5.     Structure chapter 4 & 5
- Two tables, in table 2 targets are discussed that are mentioned in table 1 (except for 1 target, see below). Chapter 5 is entitled "potential target antigens for CAR-T cells in AML"

Wouldn’t it be more logical to give another title to chapter 4 4 een andere titel te geven, eg. "Target antigen currently investigated in clinical trials"

Under chapter 4: table 1, and alinea 4.5 move to chapter 5

Chapter 5 can remin: "potential target antigens for CAR-T cells in AML"

There table 2 could be and alinea 4.5

6.     Table 1&2
No references mentioned here! Should be added!

7.     Table 1 is incomlete. Eg trial with cusatuzumab in AML (ELEVATE) NCT04150887…

8.     Table 1: maybe more info could be added, eg enrollment status, phase, recruiting sample, and last update. Eg for trial NCT04010877, the last update is from 2019 pre-covid… Is this trial still open?

9.     Table 1/2: what is the rationale behind the order? Looks randon, and it that’s indeed the case, I would use and alphabetical order

10.  Table 2: I would add the % of presence of the target Ag on AML blasts

11.  Table 2: incomplete, eg. CD38 is being investigated for AML< but not mentioned here

12.  P6 4.1. r5: instead of “GO”  Gemtuzumab ozogamicin

13.  P6 4.1. r16: “Wei-dong Han”: not mentioned in the refeernces, the author’s name is misspelled (Weidong Han), please clarify what “this construct” means

14.  P6 4.1. r17 “ Kenderian et al.”: wrong reference

15.  P6 4.1. r20 “Minagawa et al.”: year of publication is mentioned here, not for other references

16.  P6 4.1. r2: [40] is wrong  41

17.  P7 4.1. r7 : which “clinical trials” (no references)?

18.  P7 4.2. r1 “Low expressed”: english, and specify what this means.

19.  P7 4.3. R6 -> 10 “The first report… achieved CR on day 19”: is this all from the same reference? Then [56] should come at the end of the sentence.

20.  P7 4.3. r17-18 “  To overcome this issue, Lin et al. designed a PD1-silenced CLL-1 CAR and found the same cytotoxicity [58].”: This conclusion is not correct: this paper mentions “PD‑1 silencing enhanced the killing ability of CLL‑1 CAR‑T”, as illustrated in figure 3.
+ reference is wrong: 59 and not 58

21.  P8 4.4 r2 “absent in normal tissue”: not correct: CD70 is expressed on activated T, B cells and DC’s

22.  P8 4.4 r2 “highly expressed in myeloid blasts [59, 60]”: wrong reference: 62? (for sure not
58, 59, 60 or 61)

23.  P8 4.4. r6: regarding the two mentioned trials on cusatuzumab, you could mention that currently another trial is running with cusatuzumab (see remark 7)

24.  P8 4.4. r13-14 “ Based on the promising results from the combination AZA with cusatuzumab”, but could not be confirmed in phase 2: is this still promising then?

25.  P8 4.6. r7-8 “different molecules”: please specify

26.  P8 4.7 r4-5 “In two studies by Gomes-Silva et al.,”: only 1 is mentioned.

27.  P9 5. R3 “with no expression in HSCs”: should be elaborated why this is a disadvantage, because in table 2 3 potential targets are expressed on HSC's.

28.  P9 5.1. r5-8 “FRβ is an attractive target … acute promyelocytic leukemia”: reference?

29.  P9 5.1. r9 “Lynn and colleagues”, wrong reference: 76  77

30.  P9 5.1. r11->17 “They found that… limiting the off-target effect”: no reference, still Lynn et al?.

31.  P9 5.2., r1: no reference

32.  P9 5.2. r5 “TCR”: T-cell receptor

33.  P9 5.2. r5-7 “This PR1-CAR-T… normal hematopoietic progenitors.”: no reference

34.  P9 5.3. r6 “TCRm” T-cell receptor mimic

35.  P9-P10 5.3. r7 “The efficacy of this construct has been demonstrated in the lysis activity and secretion of IL-2 and IFN-. against cells line in an in vitro model.”: no reference

36.  P10 5.5. r1-> 4 “A potential new target… to design CAR-T”: no reference

37.  P10 5.5: only 1 reference?

38.  P10 5.6. r4->7 “Singlec-6 is also.. on normal HSCs”: no reference

39.  P10 5.6 r4-5 “Siglec-6 is also expressed on B-cells, mast cells and placenta and absent on on T-cells,…”  omit one ‘on’

40.  P10 5.7. r2-> 5 “The injection of TIM-3(+) AML cells into immunodeficient mice led to AML reconstitution, and these data plus the observation that TIM-3 is not expressed in normal tissues and HSCs make it an attractive target.” But how frequent is TIM-3 expression on AML?

41.   P10 5.7. r 2 -> 7 “ The injection… granzyme B, perforin.”: no reference

42.  P10 5.7 r7-> r9: Xin He et al.: ref wrong: 87  92

43.  P11 5.8.: wrong reference

44.  P11 6.1. r1 “leukemia”: only B-ALL? Or also AML? (or even CLL?)

45.  P11 6.1. r1 “ 37-93%”: very large rang  correct?

46.  P11 6.1: add CRS grading in table?

47.  P11 6.1. r15 “tisa-cel”: first mention tisagenlecleucel.

48.  P11 6.1. r17 “axi-cel”: first mention axicabtagene ciloleucel

49.  P11 6.1. r18 “brexu-cel”: first mention Brexucabtagene Autoleucel

50.  P11 6.1. r21 -> end: no reference

51.  P11 6.1. “inset”  onset

52.  P11 6.2.: title

53.  P11 6.2. r2: please specify “High-grade ICANS”, add table with grading

54.  P11 6.2. r5: please specify “severe ICANS” same as high-grade?

55.  P11 6.2. r7-8 “endothelial activation AND microglial activation”

56.  P11 6.2. r8 “ICANS management”: please also mention Anakinra in the management of ICANS, and other novel options – see review E Morris, 2022

57.  P11 6.2. r8-> 10 “ICANS management is based on the administration of corticosteroids instead the IL-6 inhibitors are often not effective for neurotoxicity associated with CAR-T cell therapy”

58.  P12 6.3. r1-3: 2 studies are mentioned, but only 1 reference

59.  P12 7. R1-2 “major limitation to CAR-T cell therapy in AML”: the mentioned limitations are general for CAR-T, not specific for AML: these should also/mainly be mentioned!

60.  P12 7.1. laatste zin “These data demonstrate the…”: not so clear how the others come to this conclusion

61.  P12 7.2.: no references!

62.  P13 8.1. r6-8 “Another study… suppressive effects on T-cells”: no reference

63.  P13 8.1. regel 13-15 “Other approaches… technology”: no reference, spelling errors

64.  P13 8.1.: please explain why armored CAR-T cells are a good approach for AML?

65.  P13 8.2. r7-11 “IL-7/CCL19-expressing…promising results”: all form one trial?

66.  P13 8.3.: no references

67.  P13 8.3. regel 3 “ADCC” =

68.  P14 8.3. “UCAR-T in AML” 8.4.

69.  P14 8.3. “UCAR-T in AML”: no references!

70.  P14 9. R5 “(LNH)” -> “(NHL)”

71.  Starting from reference 108 2 numbers

72.  Reference 109 and 110 (Cohen et al.) :2x same reference

Comments on the Quality of English Language

See other remarks

Author Response

1.No methodology: should be added.

R: thank you for pointing this out. We have added a short explanation of our methodology at the end of the introduction section.

2. References: A lot of issues, incorrect references, absent references, incorrect references (eg name author misspelled), wrong way of referring. This should be thoroughly corrected. The number of mistakes was so high, that I had to stop keeping track of them.

R: We apologize for the bibliography due to a layout error of a previous version of the bibliography in the manuscript. The corrections were inserted based on your observations.

3. The tables listing clinical trials and target antigens are incomplete. Methodology used might clarify why. …“Two tables, in table 2 targets are discussed that are mentioned in table 1 (except for 1 target, see below). Chapter 5 is entitled "potential target antigens for CAR-T cells in AML "Wouldn’t it be more logical to give another title to chapter 4 4 een andere titel te geven”… “Table 1&2 No references mentioned here! Should be added! Table 1 is incomlete. Eg trial with cusatuzumab in AML (ELEVATE) NCT04150887…Table 1: maybe more info could be added, eg enrollment status, phase, recruiting sample, and last update. Eg for trial NCT04010877, the last update is from 2019 pre-covid… Is this trial still open? Table 1/2: what is the rationale behind the order? Looks random, and it that’s indeed the case, I would use and alphabetical order Table 2: I would add the % of presence of the target Ag on AML blastsTable 2: incomplete, eg. CD38 is being investigated for AML< but not mentioned here”

R: We agree with this comment. To report the current status of the ongoing studies, in table 1 were included only the AML CAR-T trials open to recruitment. However, we double checked each trial in clinicaltrial.gov (filtered for AML CAR-T trial recruitment) and updated the table. Finally, the order of exposure of clinical trials reflects the order of the main text. With regards to target antigens in table 2, we added the percentage of blasts expression and the references, sorted in ascending numerical order.

4. Structure of part 4 & 5 is not clear and should be adapted (see below in specific comments)

R: we corrected the structure of these paragraphs.

5. Spelling mistakes should be corrected, and article should be proofread for correct English

R: Thank you for this observation; we have corrected the spelling errors and rearranged the bibliography based on your suggestions.

Reviewer 3 Report

Comments and Suggestions for Authors

the manuscript is appropriate and an interesting topic. In general it is well written andstructured but it should be more focused in AML in some sections. We recommend to the authors the following changes:

1-reduce introduction: summarize the first part of the second paragraph (from "despite"....to "elderly"); delete form "this explains" to "CBF". 

2-include a new section after the introduction that describes the research methodology used for conducting this review.

3-Tables 1 and 2 provide interesting information about clinical trials and characteristics of CART in AML. Consier including an additional table, presenting clinical information about these clinical trials: the main clinical results reported so far (efficacy and safety results), including data from posters or oral presentations at conferences. Additionally, it is recommendable to incorporate information about the types of conditioning (lymphodepleting chemotherapy) applied before CART.

4-section 4.1. delete all this (not relevant here): from "different inmunological" to "Cd16, CD123 (37,38)". Also provide clinical information regarding the clinical trials mentioned in Table 1 (i.e., target population, conditioning regimen, ORR rates (including CR), safety results).

5-section 4.2 and 4.3: comment about clinical outcomes in phase I clinical trials described in Table 1 and supported by the new table.

6-section 4.4. delete from "the history" to "reference 63".

7-section 4.5. It would be interesting to provide and comment the main clinical results (target population, ORR, safety results) of this phase 1 clinical trial (reference 67).

8-section 4.6: The information provided in this paragraph is not novel and is not of interest in relation to the study's objective, so may be omitted. Additionally, it may be confusing, as it seems that this CART is only intended for AML with mutated FLT3. It is important to clarify this and to explain the mechanism of action of this CAR-T and the target population to which it is directed, in addition to providing clinical information from the already published clinical trials.

9-Regarding section titles 4 and 5, the structure of the reading may be confusing. The CAR-Ts mentioned above are those that have already been used in clinical trials, and those described below are potential CARTs only tested in preclinical models. To enhance understanding, it would be more convenient to specify this in the section titles, for example: “CAR-T in Clinical Phase in AML” and “Potential CAR-T in Preclinical Phase in AML”.

10-section 6-could be deleted or then fully re-edited. In this section, general information is provided on the pathogenesis, clinical presentation, and management of CRS, ICANS, and post-CART cytopenias. Additionally, the results of pivotal studies for CAR-T in DLBCL are mentioned. It is suggested to omit all this information, since does not contribute novel information. Instead, section 6 could focus on the safety outcomes obtained in phase I/II clinical trials in AML: incidence, grade and management of CRS, ICANS, and post-CART cytopenias.

11-section 7. The information provided in these sections (section 7) does not offer relevant or useful information for the topic. The mechanisms of relapse or resistance to CART have been studied in myeloma and lymphoma, as mentioned in these sections, but they should not be the focus of this article. This information could be included in the context of CART in AML, if it is available.

12- section 8. The information provided in this section is very interesting (specially UCAR-T in AML), but it could be done in a more summarized way.

13-conclusions: The conclusion can be summarized. delete this part: T-cells equipped with CAR is an adaptive cell therapy that has resulted in an impressive breakthrough in the treatment of B-cell malignancies. In the recent past the FDA and EMA has approved CD19-CAR-T for treatment of R/R ALL, R/R non-Hodgkin’s lymphoma (LNH) and BCMA-CAR-T for R/R MM.

In summary, it is proposed to take a more clinical and practical approach to this review, including clinical and safety outcomes from the several AML clinical trials mentioned throughout the paper.

Author Response

1.reduce introduction: summarize the first part of the second paragraph (from "despite"....to "elderly"); delete form "this explains" to "CBF".

R: thank you for the suggestion, we corrected it

  1. Include a new section after the introduction that describes the research methodology used for conducting this review.

R: We agree with this point. At the end of Introduction, we added the Methodology.

3,4,5 Tables 1 and 2 provide interesting information about clinical trials and characteristics of CART in AML. Consider including an additional table, presenting clinical information about these clinical trials: the main clinical results reported so far (efficacy and safety results), including data from posters or oral presentations at conferences. Additionally, it is recommendable to incorporate information about the types of conditioning (lymphodepleting chemotherapy) applied before CART.

R: We followed this suggestion and added a new table (table 3) with the results of the main clinical trials in AML CAR-T. These data derived not only from scientific papers, but also in abstracts and/or oral presentation and could be fragmentary and incomplete.

6-section 4.4. delete from "the history" to "reference 63".

R: Correct according to the reviewer's suggestion

7-section 4.5. It would be interesting to provide and comment the main clinical results (target population, ORR, safety results) of this phase 1 clinical trial (reference 67).

R: We have inserted table number 3 reporting only the main clinical studies whose data reported mainly by abstracts and oral presentations are fragmentary

  1. section 4.6: The information provided in this paragraph is not novel and is not of interest in relation to the study's objective, so may be omitted. Additionally, it may be confusing, as it seems that this CART is only intended for AML with mutated FLT3. It is important to clarify this and to explain the mechanism of action of this CAR-T and the target population to which it is directed, in addition to providing clinical information from the already published clinical trials.

R: We clarify this section

  1. Regarding section titles 4 and 5, the structure of the reading may be confusing. The CAR-Ts mentioned above are those that have already been used in clinical trials, and those described below are potential CARTs only tested in preclinical models. To enhance understanding, it would be more convenient to specify this in the section titles, for example: “CAR-T in Clinical Phase in AML” and “Potential CAR-T in Preclinical Phase in AML”.

R: We agree and corrected the two-section based on the rationale CAR-T in Clinical Phase in AML” and “Potential CAR-T in Preclinical Phase in AML”.

  1. Section 6-could be deleted or then fully re-edited. In this section, general information is provided on the pathogenesis, clinical presentation, and management of CRS, ICANS, and post-CART cytopenias. Additionally, the results of pivotal studies for CAR-T in DLBCL are mentioned. It is suggested to omit all this information, since does not contribute novel information. Instead, section 6 could focus on the safety outcomes obtained in phase I/II clinical trials in AML: incidence, grade and management of CRS, ICANS, and post-CART cytopenias.

R: This suggestion is very important and we completely agree with it. We focused on AML CAR-T underlining that the same side effects showed in NHL or MM are the same in AML, even though there are scarce published data. However, in the new table 3 we added the site effects of the main clinical trials.

11-section 7. The information provided in these sections (section 7) does not offer relevant or useful information for the topic. The mechanisms of relapse or resistance to CART have been studied in myeloma and lymphoma, as mentioned in these sections, but they should not be the focus of this article. This information could be included in the context of CART in AML, if it is available.

R: We clarify this aspect before the section 7.

  1. section 8. The information provided in this section is very interesting (specially UCAR-T in AML), but it could be done in a more summarized way.

R: We agree and summarize this section. Therefore, we believed that allocarts, especially in the myeloid setting, can revolutionize the immunotherapeutic approach.

  1. conclusions: The conclusion can be summarized. delete this part: T-cells equippedwith CAR isan adaptive cell therapy that has resulted in an impressive breakthrough in the treatment of B-cell malignancies. In the recent past the FDA and EMA has approved CD19-CAR-T for treatment of R/R ALL, R/R non-Hodgkin’s lymphoma (LNH) and BCMA-CAR-T for R/R MM.

R:we agree and did it.

Round 2

Reviewer 1 Report

Comments and Suggestions for Authors

The manuscript entitled “Chimeric antigen receptor T-cell therapy in acute myeloid leukemia: state of art and recent advances” provides a comprehensive review of the current research status and progress of CAR-T therapy in acute myeloid leukemia. 

Comments on the Quality of English Language

 Minor editing of English language required

Reviewer 2 Report

Comments and Suggestions for Authors

All comments were well addressed 

Comments on the Quality of English Language

Much improved 

Reviewer 3 Report

Comments and Suggestions for Authors

no comments